**Data Availability Statement:** All relevant data are within the paper and its Supporting Information files.

# The importance of physical performance in the assessment of patients on haemodialysis: A survival analysis

**Karsten Vanden Wyngaert**[1]◉*, **Wim Van Biesen**[2]◉, **Sunny Eloot**[2]‡, **Amaryllis H. Van Craenenbroeck**[3,4]◉, **Patrick Calders**[1]◉, **Els Holvoet**[2]‡

1 Department of Rehabilitation Sciences, Faculty of Medicine and Health Sciences, Ghent University, Ghent, Belgium, 2 Department of Internal Medicine, Renal Division, Ghent University Hospital, Ghent, Belgium, 3 Laboratory of Experimental Medicine and Paediatrics, University of Antwerp, Antwerp, Belgium, 4 Department of Nephrology, University Hospitals Leuven, Leuven, Belgium

◉ These authors contributed equally to this work.
‡ SE and EH also contributed equally to this work.
* karsten.vandenwyngaert@ugent.be

## Abstract

### Background

Physical performance is an important determinant of quality of life in patients on haemodialysis. An association between physical performance and survival could further enhance the importance of physical performance. We aimed to assess the association between different measures of physical performance and survival in dialysis patients.

### Methods

117 patients on haemodialysis were included from December 2016 and followed up to September 2020. Muscle strength (quadriceps, handgrip strength, and sit-to-stand), exercise capacity (six-minute walking test, 6MWT) and the risk of falls (Dialysis Fall Index, Tinetti, and Frailty and Injuries: Cooperative Studies of Intervention Techniques) were measured at the time of inclusion. Hospitalisation, morbidity (Davies Stoke index) and death were recorded. Data were analysed by least squares linear regression models and competing risks survival hazard models.

### Results

During the observation period (median 33, min 30 max 45 months), 45 patients died (= 38.5%), resulting in a mortality rate of 15% per year. Cardiovascular disease (42.9%) was the most common cause of death. All domains of physical performance were associated with mortality, with the highest hazards for an increased risk of falls (Hazard Ratio (HR) = 20.4, $p = 0.003$) and poor exercise capacity (HR = 7.4, $p < 0.001$). A score lower than 298 meters (specificity = 0.583; sensitivity = 0.889) on the 6MWT was established as a haemodialysis-specific cut-off point for mortality risk. Each increase in 6MWT (m) corresponded with a 0.4% decrease in mortality risk (HR = 0.996, 95%CI [0.994; 0.998]). The 6MWT as also associated with comorbidity (F-value = 6.1, $p = 0.015$). Physical performance was not associated with hospitalisation.

**Funding:** The author(s) received no specific funding for this work.

**Competing interests:** The authors have declared that no competing interests exist.

## Conclusions

The 6MWT is associated with mortality in patients on haemodialysis and can be considered as a valid assessment tool to identify high-risk patients.

## Introduction

In patients with end-stage kidney disease (ESKD), substantial impairments in physical performance are common and have a considerable negative impact on quality of life (QoL) [1]. Accordingly, the vulnerable ESKD patient becomes prone to a sedentary lifestyle, leading to a vicious circle of high cardiovascular risk, frailty and further physical deterioration [2].

Physical rehabilitation programmes improve the physical performance of patients on haemodialysis and can result in a (partial) recovery of QoL [3]. Despite a well-established association between physical performance and QoL [1], it is unclear whether physical performance is also associated with survival in patients with ESKD, and which measures of physical performance have the best predictive power.

Physical performance is an umbrella term that represents the different domains involved when performing physically challenging activities such as walking or daily chores. In general, it comprises the ability to make movements by producing muscle activity (muscle strength), the capacity to perform these movements continuously for a longer period (exercise capacity), and to exert the necessary control and coordination of movements. These domains can be assessed in a purely analytical way (e.g. quadriceps peak torque at knee 90° flexed, *maximal* muscle strength) or by simulating activities of daily living (e.g. getting up from a seating position, *functional* lower limb muscle strength) [4]. The fear of falls is an important cause of a sedentary lifestyle and physical deterioration as well. An increased risk of falls has a multifactorial aetiology in patients on haemodialysis, often involving both domains of physical performance and impaired balance and coordination; and assessment of the risk of falls should therefore be included in the physical screening of these patients [5].

Various researchers aimed to identify prognostic factors that can be used in the assessment of patients with ESKD [6–8]. A measure that is associated with QoL, survival, morbidity as well as hospitalisation, could steer current assessment protocols and interventions to be more patient centred, i.e. striving for what really matters to patients rather than to theoretical constructs, contributing to a higher standard of care [9].

The aim of this study was to identify measures of physical performance that are associated with survival and hospitalisation in patients on haemodialysis. Additionally, we aimed to examine which measure(s) of physical performance was/were the most prominent in this association and could, accordingly, be relevant to be used in the clinical screening of patients on haemodialysis. We hypothesise that especially those measures which come nearest to pinpointing activities of daily living (i.e. functional measures of physical performance) are associated with long-term outcome in patients on haemodialysis.

## Materials and methods

### Participants and study design

In two tertiary dialysis centres and five satellite dialysis units, patients on haemodialysis were screened for eligibility between December 2016 and March 2018 and followed for at least 30 months. All patients were eligible except when the following exclusion criteria were present:

(1) age < 18 years, (2) pregnancy, (3) inadequate motor and verbal responses to verbal instructions and questions, and (4) < 6 months since surgical musculoskeletal interventions that could bias physical assessments.

This longitudinal study was part of a project to identify potential (predictive) biomarkers based on functional capacity, nutritional status and/or QoL in patients on dialysis (registration number on clinicaltrial.gov: NCT03910426). The present study focusses on the importance of physical performance in patients on haemodialysis to assess their survival prognosis. The results on nutritional status [10, 11], biomarkers [12] and QoL [1] are reported elsewhere. The study complies with the Declaration of Helsinki, was carried out with the approval of the local ethical committees (project number Ghent EC B670201525559; 15-OCT-2015 and Antwerp EC B300201422642; 07-DEC-2016), and written informed consents were obtained for all participants.

The sequence of physical examinations was randomized using six opaque envelopes (one for each assessment of physical function, i.e. quadriceps strength, handgrip strength, sit-to-stand test, six-minute walking test, Tinetti test, and the FICSIT) and the patients were asked to randomly order the envelopes. A minimum 3-minute interval between tests was respected. This 3-minute interval was used to let patients recover from previous examinations and to avoid bias. Patients with inabilities (e.g. wheelchair bound or amputations) were given the worst possible score for physical examinations they failed to complete. The use of walking aids was allowed except during the FICSIT test.

## Materials

Physical performance was measured by maximal and functional assessment tools that examine movements in a setting isolated from daily activities (e.g. peak quadriceps torque by isolated knee extension) and movements that are commonly performed in daily life (e.g. standing up from a seating position or walking) respectively.

**Maximal muscle strength.** Quadriceps peak torque (Microfet; Biometrics, Almere, the Netherlands) (Intraclass Correlation Coefficient (ICC) ranging between 0.76 and 0.96) [13, 14] and handgrip strength (JAMAR Hydraulic Hand Dynamometer; Patterson, Nottinghamshire, United Kingdom) (ICC = 0.93) [15] were measured by hand-held dynamometers according to the guidelines of Bohannon [16] and American Society of Hand Therapists [17] respectively. Quadriceps strength was measured by maximal isometric contraction and patients had to be in a sitting position with 90˚ flexion in both knee and hip. The contralateral arm with regard to the vascular access was assessed. Both assessments were performed in triplicate and the best effort was expressed as absolute value and as percentage of the predicted value based on age and gender [16, 18].

**Functional muscle strength.** The five repetition Sit-to-Stand test (STS) was used to examine functional lower limb muscle strength and was expressed as the time patients needed to stand up 5 times from a seating position (ICC = 0.97) [19, 20]. A cut-off value of 15 seconds was used to define impaired functional muscle strength as a longer duration has been associated with an increased risk of falls in healthy elderly [21].

**Exercise capacity.** The six-minute walking test (6MWT) is considered the gold standard for assessing functional exercise capacity and was performed following the American Thoracic Society guidelines (ICC > 0.90) [22, 23]. The test implies 6 minutes of walking in a corridor of at least 25m and was expressed as walking distance (absolute value) and as percentage to the predicted value for their gender, age, body weight and body height [24].

**The risk of falls.** The Dialysis Fall Risk Index (DFRI), as described in Vanden Wyngaert et al. [1, 5], was used and includes measures of anthropometry, inflammation, nutritional

status, maximal and functional muscle strength, exercise capacity, gait quality, and postural control as presented in S1 Table. The DFRI was scored from 0 (low risk of falls) to 12 (high risk of falls). Next, the Tinetti (ICC > 0.8) [25] and FICSIT tools (Frailty and Injuries Cooperative Studies of Intervention Technique) (ICC = 0.79) [26] were used to examine gait dysfunctions and static balance [27]. Patients scoring less than 11 on 12 on the Tinetti test were considered being at an increased risk of falls [28].

**Survival, comorbidity and hospitalisations.**   Patients' survival was calculated from the date of the physical assessments until the date of death or end of the study period (September 2020). Data on the cause of death was obtained from medical files. Transplantation during the study period was no exclusion criteria in the final survival-analyses as this would introduce censoring bias [29], they were however considered as a competing risks of survival and therefore censored via the appropriate analysis. The burden on health care was estimated by hospitalisation data of the first year after physical assessment and were obtained and expressed as the number of hospitalised days per year.

**Comorbidity.**   Comorbidity was quantified using the Davies comorbidity score, in which the following seven domains were rated on active appearance (active condition = 1; 0) [30]: (1) malignancy, (2) ischaemic heart disease, (3) peripheral vascular disease, (4) left ventricular dysfunction, (5) diabetes mellitus, (6) systemic collagen vascular disease, and (7) any other condition with a considerable negative impact on prognosis. Based on the Davies scores, patients were allocated to comorbidity grade 0 (low mortality risk, zero score), grade 1 (medium mortality risk, score 1–2) and comorbidity grade 2 (high mortality risk, score 3–7).

## Statistical analysis

Data was analysed using R statistics software version 3.5.2 (contributed libraries: *foreign*, *survival* and *cmprsk*) and SAS statistical software version 9.4 (SAS institute Inc.; Cary, NC, USA). Results are reported as mean ± standard deviation (SD), unless otherwise indicated and an alpha level of 0.05 was used.

The lower limit of normal was set on 80% of the predicted value for quadriceps strength and handgrip strength. We performed a receiver operating characteristic (ROC) analysis to identify relevant outcome measures (i.e. measures with an area under the curve (AUC) above 0.7) with regard to mortality. Subsequently, based on the Youden's index, a 298 m cut-off point (specificity = 0.583; sensitivity = 0.889) on the 6MWT test was established and used in the survival analysis (Fig 1).

We applied a competing risks survival analysis (competing event: transplantation) to examine the association between the different measures of physical performance and mortality as described by Fine and Gray [31]. Two models were used in the adjusted survival analysis: (1) model 1 included age and gender as confounding factors and (2) model 2 added the Davies morbidity index and dialysis vintage to the factors of model 1. Also, the unique associations between measures of physical performance and survival were analysed via a univariate analysis. A multivariate regression model was used to examine which measure(s) of physical performance was/were most prominent in the association between physical performance and survival. Next, least squares linear regression models were used to evaluate the association between hospitalisation and comorbidity on the one hand and the different measures of physical performance on the other. In order to increase statistical power, only significant variables in the unadjusted multivariate analysis (p<0.05) were included in the adjusted analyses. The DFRI was excluded from the multivariate analyses due to collinearity (variance inflation factor > 4).

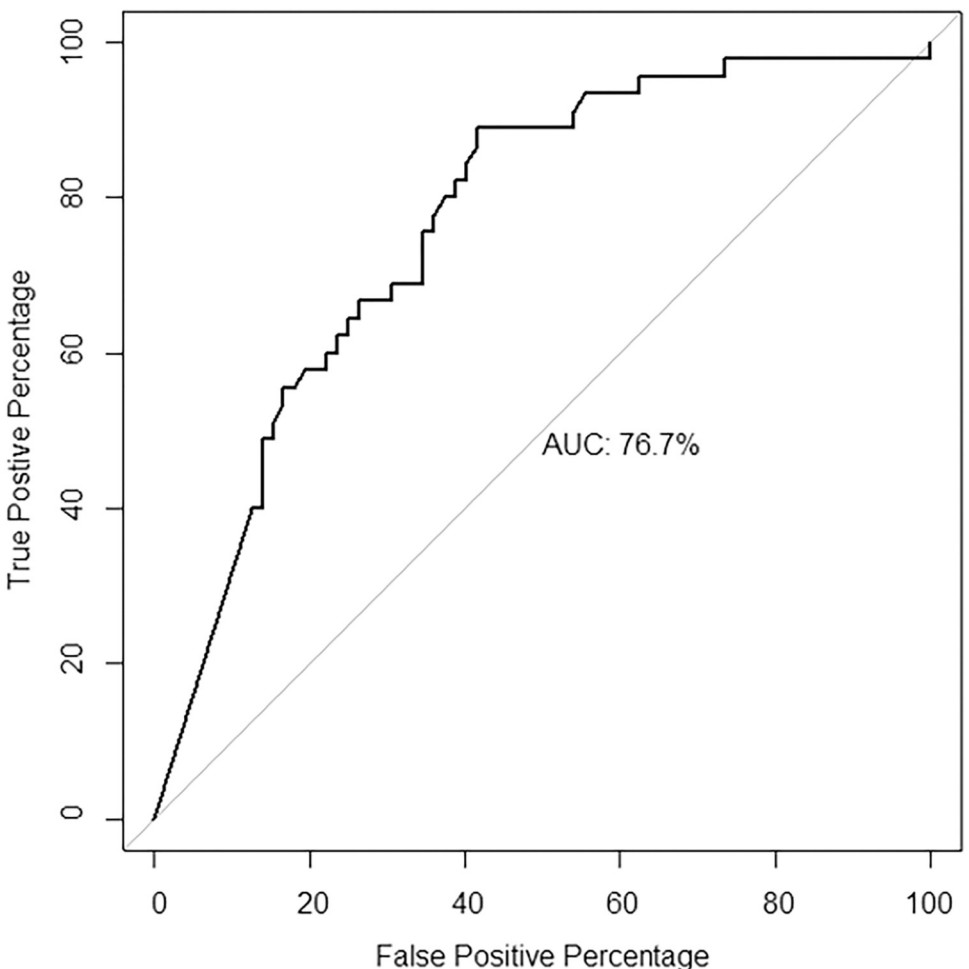

**Fig 1. ROC-analysis by 6MWT.**

## Results

### Demographics

This study included 117 (low-/high-care centres n = 54/n = 63) patients on maintenance hae-modialysis (average 68 years ± 16; 41% female) and obtained data on their physical perfor-mance, mortality, comorbidity, and hospitalisation. The cohort is representative for the dialysis population in Flanders [32] and the main patient characteristics are reported in Table 1. During an observation period of 33 months on average (ranging between 30 and 45 months), 45 subjects (38.5%) died and 13 patients (11.1%) were transplanted, which resulted in a mortality rate of 15% and rate of censoring of 18%. No patients dropped out during the observation period. Sixteen, fifteen, six, and eight patients died in the first, second, third, and fourth annum, which corresponds with a relative ratio of 35.6%, 33.3%, 13.3%, and 17.8%. With respect to the competing risks, six, four, and three patients were transplanted in the first, second and third annum respectively, resulting in a medium censoring time of 425 days, [min-imum 49 days; maximum 796 days]. The causes of death were cardiovascular disease (42.9%), cachexia (26.2%), infection (9.5%), and others (21.4%; including cancer, liver failure, and unknown causes). The median dialysis vintage was 3 years, ranging between 0 and 16 years. In general, the majority of patients showed impairments in all measures of physical performance

**Table 1. Patient characteristics.**

| Variable | n = 117 |
|---|---|
| Age (years) | 68 ± 16.0 |
| Female sex | 48 (41.0) |
| BMI (kg/m2) | 26 ± 5.3 |
| Quadriceps strength (Newton) | 178 ± 75.5 |
| % of predicted value | 53.3 ± 17.9 |
| Patients with pathological value (n; %) | 108; 92.3 |
| Handgrip strength (kg) | 28 ± 11.3 |
| % of predicted value | 90.1 ± 31.2 |
| Patients with pathological value (n; %) | 43; 36.8 |
| Sit-to-Stand (seconds) | 30 ± 17.7 |
| Patients at increased risk of falls (n; %) | 81; 69.2 |
| 6MWT (m) | 240 ± 192.9 |
| % of predicted value | 40 ± 30.0 |
| Patients scoring <298m (n; %) | 80; 68.4 |
| Dialysis fall index (0–12) | 6 ± 3.1 |
| Patients at increased risk of falls (n; %) | 87; 74.4 |
| Tinetti (0–12) | 8 ± 4.0 |
| Patients at increased risk of falls (n; %) | 58; 49.6 |
| FICSIT (0–28) | 14 ± 8.7 |
| Dialysis vintage (months) | 36 [11; 55] |
| Comorbidities (n; %) | |
| Diabetes | 53; 45.7 |
| CVD | 86; 73.5 |
| Neuropathy | 32; 27.4 |
| Retinopathy | 36; 30.8 |
| Respiratory complications | 30; 25.6 |
| Musculoskeletal complications | 53; 45.3 |
| Davies comorbidity score (0–7) | 2 [1; 3] |
| Daily drug use | 13.6 ± 3.6 |
| <10 drugs (n; %) | 14; 12.0 |
| 10–14 drugs (n; %) | 56; 47.9 |
| ≥15 drugs (n; %) | 47; 40.2 |

Data are presented as mean ± standard deviation, as number (percentage) and as mean [$25^{th}$; $75^{th}$] quantiles. BMI, body mass index; CVD, cardiovascular disease; VAS, visual analogue scale; 6MWT, 6-minute walking test

and an increased risk of falls. Only 9% of the patients scored levels of exercise capacity expected levels for their age and sex.

## Physical performance and mortality

All domains of physical performance were associated with survival in the univariate competing risks regression models (Tables 2 and 3). Although quadriceps peak torque was not associated with mortality risk, patients with an established quadriceps muscle weakness for that patients' age and sex were at higher risk for death compared to those without (HR = 5.291, $p = 0.045$). The risk of falls by DFRI was associated with an increased mortality risk (HR = 20.4, $p = 0.003$). A walking distance of less than 298 meters (specificity = 0.583; sensitivity = 0.889, Fig 1) was associated with an increased mortality risk (HR = 7.4, $p = 0.045$). Only exercise

**Table 2. Unadjusted competing risks regression model for mortality.**

| | Mortality (event = death) | | | |
|---|---|---|---|---|
| | Univariate model | | Multivariate model | |
| | HR | p-value | HR | p-value |
| Quadriceps strength (N) | 1.000 | 0.665 | / | NA |
| Handgrip strength (kg) | 0.965 | 0.004 | 0.998 | 0.918 |
| Sit-to-Stand (s) | 1.000 | <0.001 | 1.000 | 0.407 |
| FICSIT (/28) | 1.000 | <0.001 | 0.954 | 0.098 |
| Tinetti (/12) | 1.000 | <0.001 | 1.048 | 0.098 |
| DFRI (/12) | 1.352 | <0.001 | / | NA |
| 6MWT (m) | 0.996 | <0.001 | 0.997 | 0.005 |

Data are presented as Hazard Ratios (HR). Introduced factors in the multivariate model were handgrip strength, Sit-to-Stand, FICSIT, Tinetti, and 6MWT. Multivariate model p<0.001 and Chi-square = 47.3585. DFRI, dialysis fall risk index; 6MWT, six-minute walking test

capacity as a continuous variable was retained as a predictor of survival in the multivariate analyses (6MWT (m): HR = 0.997, $p$ = 0.005; impaired 6MWT: HR = 5.880, $p$ = 0.002). After adjusting for age, sex, comorbidity and dialysis vintage, each increase on the 6MWT (m) was associated with a 0.4% decrease in mortality risk (HR = 0.996, 95%CI [0.994; 0.998], $p$<0.001, Table 4). The results of the adjusted survival analysis for the 298 meters walking distance cut-off point (i.e. the categorical analysis of exercise capacity) were similar to the results of the unadjusted analysis (HR = 7.576, $p$<0.001, Table 5).

## Physical performance, comorbidity and hospitalisations

Based on the Davies comorbidity index, 16.2%, 49.6%, and 34.2% of the population were identified as having a low, medium, and high mortality risk. Cardiovascular disease was the most prevalent comorbidity (73.5%), followed by Type 2 Diabetes Mellitus (45.7%), and musculo-skeletal complications (45.3%) of which gout was the predominant cause (21.4%). Polypharmacy (≥ 5 different drugs per day) and excessive drug use (≥ 10 drugs per day) were highly prevalent (95.6% and 88.0% of the cohort respectively), reflecting the high degree of comorbidity [33]. Similar to the survival analysis, all domains of physical performance were associated with the degree of morbidity based on the Davies comorbidity scale (0–7), albeit only the

**Table 3. Unadjusted competing risks regression model for mortality by the presence of impairments in physical performance.**

| | Mortality (event = death) | | | |
|---|---|---|---|---|
| Impairments (impaired (n); not impaired (n)) | Univariate model | | Multivariate model | |
| | HR | p-value | HR | p-value |
| Quadriceps strength (108; 9) | 5.291 | 0.045 | 2.611 | 0.293 |
| Handgrip strength (43; 74) | 2.433 | 0.003 | 1.721 | 0.079 |
| Sit-to-Stand (81; 36) | 4.566 | 0.001 | 1.403 | 0.573 |
| Tinetti (58; 59) | 3.215 | <0.001 | 0.001 | 0.778 |
| DFRI (87; 30) | 20.408 | 0.003 | / | NA |
| 6MWT (80; 37) | 7.407 | <0.001 | 5.882 | 0.002 |

Data are presented as Hazard Ratios (HR). Introduced factors in the multivariate model were quadriceps strength, handgrip strength, Sit-to-Stand, Tinetti, and 6MWT. Multivariate model p<0.001 and Chi-square = 47.1703. DFRI, dialysis fall risk index; 6MWT, six-minute walking test

**Table 4. Adjusted competing risks regression model for mortality.**

|  | Model 1 | | Model 2 | |
|---|---|---|---|---|
|  | HR | p-value | HR | p-value |
| 6MWT (m) | 0.996 | <0.001 | 0.996 | <0.001 |

Data are presented as Hazard Ratios (HR). Model 1 was adjusted for age and sex; model 2 for age, sex, Davies morbidity index, and dialysis vintage. Model 1: p<0.001 and Chi-square = 27.2982; Model 2: p<0.001 and Chi-square = 37.0895. 6MWT, six-minute walking test

**Table 5. Adjusted multivariate competing risks regression model by the presence of impairments in physical performance.**

|  | Model 1 | | Model 2 | |
|---|---|---|---|---|
|  | HR | p-value | HR | p-value |
| 6MWT | 7.353 | <0.001 | 7.576 | <0.001 |

Data are presented as Hazard Ratios (HR). Model 1 was adjusted for age and sex; model 2 for age, sex, Davies morbidity index, and dialysis vintage. Model 1: p<0.0001 and Chi-square = 25.7888; Model 2: p<0.001 and Chi-square = 31.1347. 6MWT, six-minute walking test

**Table 6. Unadjusted least squares linear regression model of the different measures of physical performance on comorbidity (Davies comorbidity scale 0–7).**

|  | Davies comorbidity index | | | |
|---|---|---|---|---|
|  | Univariate model | | Multivariate model | |
|  | F-value | p-value | F-value | p-value |
| Quadriceps strength (N) | 6.415 | 0.013 | 0.808 | 0.371 |
| Handgrip strength (kg) | 3.804 | 0.050 | 0.999 | 0.320 |
| Sit-to-Stand (s) | 8.632 | 0.004 | 0.000 | 0.992 |
| FICSIT (/28) | 8.200 | 0.005 | 0.151 | 0.698 |
| Tinetti (/12) | 6.889 | 0.010 | 0.123 | 0.726 |
| DFRI (/12) | 29.364 | <0.001 | / | NA |
| 6MWT (m) | 14.009 | <0.001 | 4.549 | 0.035 |

Data are presented as F-values. Multivariate model p = 0.012 and R-squared = 0.145. DFRI, dialysis fall risk index; 6MWT, six-minute walking test

**Table 7. Adjusted least squares linear regression model of the 6MWT on comorbidity (Davies comorbidity scale 0–7).**

|  | Model 1 | | Model 2 | |
|---|---|---|---|---|
|  | F-value | p-value | F-value | p-value |
| 6MWT (m) | 6.111 | 0.015 | 4.365 | 0.039 |

Data are presented as F-values. Model 1 was adjusted for age and sex; model 2 for age, sex, and dialysis vintage. Model 1: p<0.001 and R-squared = 0.206. Model 2: p<0.001 and Chi-square = 0.208. 6MWT, six-minute walking test

6MWT remained relevant in the multivariate analysis ($R^2$ = 0.145, $p$ = 0.012, Table 6). Adjusted for age, sex and dialysis vintage, the 6MWT explained 20.8% of the variance in the Davies comorbidity scale ($p$<0.001, Table 7).

**Table 8. Least squares linear regression model of the different measures of physical performance on 1-year hospitalisation (days/year).**

| | Unadjusted univariate model | | Adjusted univariate model 1 | | Adjusted univariate model 2 | |
|---|---|---|---|---|---|---|
| | F-value | p-value | F-value | p-value | F-value | p-value |
| Quadriceps strength (N) | 0.371 | 0.544 | 0.898 | 0.346 | 0.357 | 0.551 |
| Handgrip strength (kg) | 0.047 | 0.829 | 0.498 | 0.482 | 0.195 | 0.660 |
| Sit-to-Stand (s) | 0.001 | 0.971 | 0.010 | 0.922 | 0.097 | 0.757 |
| FICSIT (/28) | 1.057 | 0.306 | 0.879 | 0.352 | 1.023 | 0.314 |
| Tinetti (/12) | 0.879 | 0.350 | 1.215 | 0.273 | 1.698 | 0.195 |
| DFRI (/12) | 3.768 | 0.055 | 4.416 | 0.038 | 2.420 | 0.123 |
| 6MWT (m) | 0.787 | 0.377 | 0.920 | 0.339 | 0.460 | 0.499 |

Data are presented as F-values. Model 1 was adjusted for age and sex; model 2 for age, sex, Davies comorbidity index and dialysis vintage. DFRI, dialysis fall risk index; 6MWT, six-minute walking test

Notwithstanding a trend towards an association between the risk of falls by DFRI and the number of admitted days to the hospital, no measures of physical performance determined the 1-year hospitalisation data (Table 8).

## Discussion

This prospective cohort study demonstrates that the 6MWT is a useful tool in the assessment and screening of patients on maintenance haemodialysis. We report baseline data of 117 patients and followed them for at least 30 months. Forty-five deaths were noted, and 13 patients received a kidney graft. The main cause of death was cardiovascular disease. Although all measures of physical performance (i.e. muscle strength, exercise capacity, and the risk of falls) were associated with mortality, only exercise capacity by 6MWT was identified as an important determinant in a multivariate analysis. The present study is the first to indicate that a walking distance of 298-meters on the 6MWT could be used as cut-off point indicative of prognosis. This cut-off point remained robust after adjustment for age, sex, comorbidity and dialysis vintage. No measures of physical performance were related to 1-year hospitalisation.

There is a vicious circle between cardiovascular morbidity and physical inactivity [34]. Poor physical performance perpetuates this circle, being both cause and effect of sedentary behaviour. Impairments in measures of physical performance representative for activities of daily living are most suitable to assess risk for entering this downward spiral. So, in the case of lower limb muscle strength, the ability to stand up from a seated position would contribute more to the above reported vicious circle than maximal quadriceps strength, as is confirmed by the higher association of functional muscle strength (by STS) with cardiovascular morbidity compared to maximal muscle strength (by quadriceps peak torque as well as handgrip strength) observed in our cohort. As cardiovascular events occur in 75% of our cohort, a higher predictive power of the STS compared to quadriceps peak torque for mortality comes as no surprise as well.

Cardiovascular disease is the leading comorbid disease and accounts for approximately 50% of all-cause mortality in patients with ESKD [35]. A measure of physical performance that reflects cardiovascular health could therefore be of critical importance in the risk assessment of these patients. Peak oxygen uptake (VO$_2$peak) is measured during a maximal cardiopulmonary exercise test and is associated with cardiorespiratory function and mortality in most severe chronic diseases such as heart failure, obstructive and restrictive lung diseases and in transplant recipients [36–39]. This maximal and non-functional measure of exercise capacity is closely associated with the 6MWT [40]. Although VO$_2$peak is considered the gold standard

for measuring cardiovascular health, the 6MWT was reported to be superior to VO$_2$peak as a risk stratification tool in dialysis patients. Also, the assessment of VO$_2$peak is expensive, labour intensive and time-consuming and requires special know-how on sports physiology. Furthermore, a basic motor skill is expected of the patient. For all these reasons, cardiopulmonary exercise testing is not realistic as a screening and assessment tool in the standard care of patients on haemodialysis [41].

The mortality-threshold of approximately 300 meters walking distance on the 6MWT derived in our cohort is consistent with data from other chronic populations (ranging between 240 and 350 meters) [42–44]. The 4% decrease in mortality risk for each increase of 10 meters in the present study is in line with a study by Kohl et al., reporting a decrease of 47% per 100 meter increments [40]. Yet, our results indicate that this association cannot be fully explained by cardiovascular disease alone. Other components than cardiovascular function can impact the 6MWT, some of which contribute to the association between 6MWT and mortality as well.

First, *frailty* is common in the dialysis-dependent population and is defined as a decreased physiological reserve and resilience to stressors, resulting in physical deterioration, disability, an increased burden on healthcare, and eventually death [45]. The Fried's frailty checklist is the most commonly cited assessment tool and includes the following criteria: unintentional weight loss, self-reported exhaustion, decreased muscle strength, slow walking speed, and low physical activity [46, 47]. All these parameters are independently associated with mortality, with gait speed being the strongest predictor of survival in patients on haemodialysis [48, 49]. In line with Johansen et al., [49] we found that protein-energy wasting (i.e. cause and effect of frailty) is a major determinant of 6MWT as well [11].

Second, after CV disease and infections, withdrawal from dialysis due to cachexia is the third most common cause of death (15–25% of all-cause mortality) in patients on haemodialysis in the United States [35], in which the *quality of life* is a major decisive component [50]. As reported in our previous study, the 6MWT contributed substantially to impaired health-related QoL and health utility in patients on haemodialysis [1]. Whereas no association was reported between cardiovascular disease and withdrawal of dialysis in a review by Qazi et al., [50] we posit that the association between the 6MWT and QoL could contribute to the 6MWT's greater prognostic value, albeit especially in patients where low health-related QoL has a direct impact on mortality.

Based on the different associations between 6MWT on the one hand and quality of life, nutritional status, mortality and morbidity on the other, we state that the 6MWT is a reliable and patient-relevant screening tool in patients on haemodialysis, that could contribute to a higher standard of care and better follow-up in this population.

Next to exercise capacity, the risk of falls was identified as a determinant of survival and morbidity in our population as well. Falls have been associated with a 2.5 higher hospitalisation rate and a 1.5 higher mortality rate [51]. However, the present study found that an increased risk of falls, as based on a screening tool tailored to haemodialysis patients, is associated with a 20-fold higher mortality risk compared to patients with low risk of falls. Despite the magnitude of this result, this association was not unexpected as the DFRI includes a wide range of variables which are directly related to mortality such as age, protein-energy wasting, frailty, exercise capacity, and cardiovascular responsiveness to dialysis-related hypotension. Therefore, it is reasonable to state that a tailored-to-dialysis risk of falls assessment tool might be a better predictor for mortality than the number of accidental falls.

The first and main limitation of this study was that the number of actual accidental falls was not registered in the follow-up period. This absolute number was estimated using a validated risk of falls screening tool tailored to dialysis patients. Despite respecting the item reliability, some adaptations had to be made in the DFRI as described in Vanden Wyngaert et al. [1].

Second, although we excluded variables that caused multicollinearity into the statistical models, some collinearity between the remaining measures of physical function may still be present. Third, another limitation was that no maximal measures of exercise capacity (e.g. VO$_2$peak) were included. Nevertheless, this study was the first of its kind to examine the association of both absolute and functional measures of physical performance with mortality in a haemodialysis cohort. Fourth, mortality data were used to calculate both a cut-off point and hazard ratio's, potentially bringing some bias to the analysis. In addition, bias due to censoring might still be present despite the use of a competing risks analysis. Therefore, our data should be interpreted with caution. Fifth, the variable selection in the multivariate analysis was based on the univariate analysis and not on a stepwise selection model. Together with previously published data, this study provides a comprehensive discussion and analysis of different measures of physical performance and their association with patient-relevant outcome measures. As such, based on this study, we recommend clinicians and researchers to include the 6MWT in their assessment protocols.

## Conclusions and guidelines for further research

In conclusion, functional measures of physical performance are associated with mortality and morbidity in patients on haemodialysis. The 6MWT is a relevant and useful tool in the screening and should be included in standard care assessment protocols of these patients. In order to use the 6MWT as a screening tool, future research should aim to establishing the minimal clinically important difference in performance on this test in patients on haemodialysis. Additionally future research should try to validate and examine the cut-off point in other cohorts of haemodialysis patients as well as to examine the true clinical importance of the associations reported in the present study.

## Supporting information

**S1 Table. Dialysis fall risk index.**
(DOCX)

**S1 Dataset. Anonymized dataset.**
(XLSX)

## Acknowledgments

The authors are indebted to IVD and SV for database management, and to the study nurses EDM and KR for assistance with the questionnaires.

## Author Contributions

**Conceptualization:** Karsten Vanden Wyngaert, Wim Van Biesen, Sunny Eloot, Amaryllis H. Van Craenenbroeck, Els Holvoet.

**Formal analysis:** Els Holvoet.

**Methodology:** Karsten Vanden Wyngaert, Wim Van Biesen, Sunny Eloot, Amaryllis H. Van Craenenbroeck, Els Holvoet.

**Supervision:** Karsten Vanden Wyngaert, Amaryllis H. Van Craenenbroeck, Patrick Calders, Els Holvoet.

**Visualization:** Sunny Eloot.

Writing – original draft: Karsten Vanden Wyngaert, Amaryllis H. Van Craenenbroeck, Els Holvoet.

Writing – review & editing: Wim Van Biesen, Sunny Eloot, Amaryllis H. Van Craenenbroeck, Patrick Calders, Els Holvoet.

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
