## [Decision Letter · Decision Letter 0]

6 May 2021

PONE-D-20-37730

The importance of physical performance in the assessment of patients on haemodialysis: a survival analysis

PLOS ONE

Dear Dr. Vanden Wyngaert,

Thank you for submitting your manuscript to PLOS ONE. After careful consideration, we feel that it has merit but does not fully meet PLOS ONE’s publication criteria as it currently stands. Therefore, we invite you to submit a revised version of the manuscript that addresses the points raised during the review process.

Please carefully review the comments provided by the reviewers, especially as they related to point 3 of PLOS ONE’s publication criteria.

We look forward to receiving your revised manuscript.

Kind regards,

Melissa M Markofski

Academic Editor

PLOS ONE

Journal Requirements:

Reviewers' comments:

Reviewer's Responses to Questions

**Comments to the Author**

1. Is the manuscript technically sound, and do the data support the conclusions?

Reviewer #1: Partly

Reviewer #2: Yes

2. Has the statistical analysis been performed appropriately and rigorously? 

Reviewer #1: No

Reviewer #2: Yes

3. Have the authors made all data underlying the findings in their manuscript fully available?

Reviewer #1: Yes

Reviewer #2: Yes

4. Is the manuscript presented in an intelligible fashion and written in standard English?

Reviewer #1: No

Reviewer #2: Yes

5. Review Comments to the Author

Reviewer #1: This paper presents a reasonable, if incomplete, analysis of a set of predictor markers for quality of life and survival. Overall many of the methods are sounds and of reasonable sophistication. However, major concerns still remain.

In assessing categorical outcomes (survival or hospitalization, in particular) it is generally preferred to fit a model based on a benchmark time point (i.e., 1 year). Additionally, inferential results (p¬-values, confidence intervals, etc.) are listed in a haphazard manner among the results, with some inferential findings missing completely. Finally, the purpose of the study should be listed more clearly. The use of the 6MWT assessment tool is justified, but the multi-variate modeling procedure is incomplete and missing useful details about the prediction of outcomes. This is especially important given the disparity between significance in the univariate and multivariate analyses.

More specific comments are given below:

1. Line 30: The 15/100 patient deaths per year is plausible but should either be presented as a percentage or in terms of the total number of patient-years.

2. Line 33-36: It is not obvious why there are no inferential descriptors (p-values, confidence intervals) in this set of results.

3. Line 78: No description is found about the effect of center on the results. At the least, the number of patients contributed by each center should be given.

4. Line 93: If the sequence of physical examinations was indeed randomized, then more detail is required. Discussion of how the randomization was done and the relative effect of the random order would be important.

5. Line 134: This sentence is not clearly written. It appears that you did not exclude patients who received transplants.

6. Line 148: The contributed libraries used in this analysis should also be listed.

7. Lines 153-157: It appears that the ROC curve was used to assess a cutoff for using 6MWT as a predictor of mortality. It is not clear how the area under the curve was instrumental, unless other models or predictors were used, in which case they should be described. Given the ROC curve described, it is not clear, based on the description, how the cutoff point was chosen.

8. Line 166: It appears that the authors mean “generalized linear models”.

9. Line 168: Stepwise methods or the variety of penalized regression modeling techniques are preferable modeling approaches to pulling out significant variables in a univariate analysis.

10. Line 169: Based on the results (comparing inferences in the univariate and multivariate models), it appears that more collinearity was a bigger issue than reported. The results do not make clear why the results are so divergent.

11. Lines 196/198: The change in mortality risk should be given in terms of the statistics they were based upon.

12. Lines 200/202: The significance of 6MWT in this setting is surprising and difficult to believe, given how close it is to one. Standard errors or confidence intervals might help here.

13. Line 202: It is not clear how a hazard ratio of 0.996 translates into a 4% change in mortality risk.

14. Line 276-277: The robustness of the cut-off is not justified, since there is no obvious attempt to assess the factor 6MWT as it interacts with the covariates listed.

Reviewer #2: Overall, the idea of research is very interesting to be studied nowadays and paper is coherently developed. However, there are some comments and suggestions.

Abstract

Page 2 line 21: add ''Aim'' before (To assess the association between different 22 measures of physical performance and survival in dialysis patients).

Page 2 line 26: write the full name of (FICSIT-4)

Page 2 line 32: write the full name of (HR)

Keywords: write it in alphabetical order

Introduction

Line 53 to 64, please insert the reference for this section

Materials and methods

Study design is not mentioned

Correct the title on page 4 line 76 (Materials an& methods) to (Materials and methods)

Page 5 line 103: please clarify the (Quadriceps peak torque) test protocol as did you examine eccentric or concentric Quadriceps peak torque, the knee at angular velocity

Validity and reliability of the used instruments used in this study should be clarified

Statistical analysis

Page 7 line 163 the authors mentioned the reference Fine and Gray [23, 24]. However 23 and are two different references. Correct

Delete the word abbreviation bellow all tables

Funding

Page 17 line 372 and 373 include funding and conflicts of interest. Please write in separate

References

Update some references is required

6. PLOS authors have the option to publish the peer review history of their article (what does this mean?). If published, this will include your full peer review and any attached files.

Reviewer #1: No

Reviewer #2: No

---

## [Author Response · Author response to Decision Letter 0]

1 Nov 2021

REVIEWER #1

This paper presents a reasonable, if incomplete, analysis of a set of predictor markers for quality of life and survival. Overall many of the methods are sounds and of reasonable sophistication. However, major concerns still remain.

We thank the reviewer for his/her constructive criticism and interest in our work.

1) In assessing categorical outcomes (survival or hospitalization, in particular) it is generally preferred to fit a model based on a benchmark time point (i.e., 1 year). 

Indeed, a benchmark of one year is an often used time point when examining categorical outcomes (such as incidence data). Nevertheless, when performing a survival analysis, a time-to-event analysis can be preferred because it results in more reliable effect sizes. Also, due to the relatively small sample size, the number of events – and thereby the power and appropriateness to use covariates in the analysis such as age and sex– was too low when considering the cut-off point of 1 year. Accordingly, we preferred to report a time-to-event analysis, which method has been frequently used in literature as well (Ebrahimi et al., Factors influencing survival time of hemodialysis patients; time to event analysis using parametric models: a cohort study. BMC Nephrol 20, 215 & Zheng C, Tian J, Wang K, Han L, Yang H, Ren J, Li C, Zhang Q, Han Q, Zhang Y. Time-to-event prediction analysis of patients with chronic heart failure comorbid with atrial fibrillation: a LightGBM model. BMC Cardiovasc Disord. 2021 Aug 4;21(1):379.). 

2) Additionally, inferential results (p-values, confidence intervals, etc.) are listed in a haphazard manner among the results, with some inferential findings missing completely. 

We thank the reviewer for this remark, which will improve the transparency of our manuscript and agree that the reporting of inferential results could and should be improved in our manuscript. The following inferential results were added and adapted to the body of the results section of the revised version of the manuscript. 

Results, Page 10, lines 195-197: “This study included 117 (low-/high-care centres n=54/n=63) patients on maintenance haemodialysis (average 68 years ± 16; 41% female) and obtained data on their physical performance, mortality, comorbidity, and hospitalisation.”

Results, Page 11, line 218-222: “The risk of falls by DFRI was associated with an increased mortality risk (HR = 20.4, p=0.003). A walking distance of less than 298 meters (specificity = 0.583; sensitivity = 0.889, Fig. 1) was associated with an increased mortality risk (HR = 7.4, p=0.045).”

Results, Page 11, line 224-226: “After adjusting for age, sex, comorbidity and dialysis vintage, each increase on the 6MWT (m) was associated with a 0.4% decrease in mortality risk (HR = 0.996, 95%CI [0.994; 0.998], p<0.001, Table 4).”

3) Finally, the purpose of the study should be listed more clearly. The use of the 6MWT assessment tool is justified, but the multi-variate modelling procedure is incomplete and missing useful details about the prediction of outcomes. This is especially important given the disparity between significance in the univariate and multivariate analyses.

Thank you for this comment. We adapted the aim of the study in the revised version of the introduction to make it more transparent. The main aim of this study was to examine which measures of physical performance are related to survival and hospitalisation in patients on haemodialysis. This research question was answered by the univariate analysis. Additionally, we also aimed to identify the most prominent measure of physical performance. This second research question was analysed via a backward stepwise regression (multivariate) model. Whereas these two aims were not clearly reported, we adapted the following sentences:

Introduction, page 5, lines 75-80: “The aim of this study was to identify measures of physical performance that are associated with survival and hospitalisation in patients on haemodialysis. Additionally, we aimed to examine which measure(s) of physical performance was/were the most prominent in this association and could, accordingly, be relevant to be used in the clinical screening of patients on haemodialysis.”

Additionally, we added an explanation of which statistical methods were used to answer the main and secondary research question. The following sentences were added to the revised version of the Statistical Analysis:

Materials and methods, page 9, lines 182-186: “Also, the unique associations between measures of physical performance and survival were analysed via a univariate analysis. A multivariate stepwise backward regression model was used to examine which measure(s) of physical performance was/were most prominent in the association between physical performance and survival.”

More specific comments are given below:

4a) Line 30: The 15/100 patient deaths per year is plausible but should either be presented as a percentage or in terms of the total number of patient-years.

Thank you for this suggestion. We adapted the sentence.

The following sentences were adapted in the revised version of the manuscript

Abstract, page 2, lines 30-31: “During the observation period (median 33, min 30 max 45 months), 45 patients died (=38.5%), resulting in a mortality rate of 15% per year.”

Results, page 10, line 200-201: “This resulted in a mortality rate of 15%.”

4b) Line 33-36: It is not obvious why there are no inferential descriptors (p-values, confidence intervals) in this set of results.

We agree with the reviewer that some important inferential descriptors were lacking in the abstract. We also agree that this information is important for the readers to interpret the abstract. Accordingly, we added information on the inferential descriptors to the abstract.

The following sentences were adapted and added to the revised version of the abstract:

Abstract, page 2, lines 34-40: “A score lower than 298 meters (specificity = 0.583; sensitivity = 0.889) on the 6MWT was established as a haemodialysis-specific cut-off point for mortality risk. Each increase in 6MWT (m) corresponded with a 0.4% decrease in mortality risk (HR = 0.996, 95%CI [0.994; 0.998]). The 6MWT as also associated with comorbidity (F-value = 6.1, p=0.015).”

4c) Line 78: No description is found about the effect of centre on the results. At the least, the number of patients contributed by each centre should be given.

Thank you for this remark. Patients were recruited from 2 high-care dialysis units (Ghent n=41 and Antwerp n=22 patients) and 5 low-care dialysis units (Ghent n=10 ; Aalst n=19 ; Geraardsbergen n=5 ; Wetteren n=7 ; and Antwerp n=13). Note that the dialysis units in Aalst, Geraardsbergen, and Wetteren are satellite units of Ghent University Hospital, resulting in an Antwerp/Ghent recruitment-ratio of 35/82. Data regarding the selection process and the high-care/low-care ratio (i.e. 63/54) was added to the revised version of the manuscript. 

Since Antwerp and Ghent are cities that are a 35-minutes’ drive away from each other, and no differences in standard care or representative cohort characteristics are expected, we decided to report the number of patients recruited from low-care versus high-care dialysis centres, because this information is an important characteristic of our cohort.

The following information was added to the revised version of the manuscript: 

Results, page 10, lines 195-197: “This study included 117 (low-/high-care centres n=54/n=63) patients on maintenance haemodialysis (average 68 years ± 16; 41% female) and obtained data on their physical performance, mortality, comorbidity, and hospitalisation.”

4d) Line 93: If the sequence of physical examinations was indeed randomized, then more detail is required. Discussion of how the randomization was done and the relative effect of the random order would be important.

We used opaque envelopes to randomise the sequence of examinations. The requested information was added to the manuscript.

The following information was added to the revised version of the manuscript:

Methods, page 6, lines 101-103: “The sequence of physical examinations was randomized using opaque envelopes and a minimum 3-minute interval between tests was respected. This 3-minute interval was used to let patients recover from previous examinations and to avoid bias.”

4e) Line 134: This sentence is not clearly written. It appears that you did not exclude patients who received transplants.

Thank you. Patients that were transplanted during the follow-up time were censored, via the appropriate analysis (i.e. Competing Risk). They were not excluded from the baseline data as this would induce a selection bias. Also patients that had a failed transplant were censored. I adapted the sentence to improve the transparency of this study.

The following information was added to the revised version of the manuscript:

Methods, page 8, lines 148-150: “Transplantation during the study period was no exclusion criteria in the final survival-analyses as this would introduce censoring bias [29], they were however considered as a competing risk of survival and therefore censored via the appropriate analysis.”

4f) Line 148: The contributed libraries used in this analysis should also be listed.

Thank you for this suggestion. The libraries were added to the Statistical Analysis.

The following information was added:

Material and methods, page 8, lines 163-165: “Data was analysed using R statistics software version 3.5.2 (contributed libraries: foreign, survival and cmprsk) and SAS statistical software version 9.4 (SAS institute Inc.; Cary, NC, USA).”

4g) Lines 153-157: It appears that the ROC curve was used to assess a cut-off for using 6MWT as a predictor of mortality. It is not clear how the area under the curve was instrumental, unless other models or predictors were used, in which case they should be described. Given the ROC curve described, it is not clear, based on the description, how the cut-off point was chosen.

Thank you for this suggestion to improve the transparency of our manuscript. We used the Youden’s Index to calculate the cut-off point on the ROC. This point is the maximum point of = sensitivity+specificity – 1. The area under the curve was only calculated to identify models that were clinically relevant. 

The following information was added to the revised version of the manuscript:

Materials and methods, page 9, lines 172-174: “Accordingly, based on the Youden’s index, a 298 m cut-off point (specificity = 0.583; sensitivity = 0.889) on the 6MWT test was established and used in the survival analysis (Fig. 1).”

4h) Line 166: It appears that the authors mean “generalized linear models”.

Yes, we indeed wanted to refer to generalized lineal models. Thank you for addressing this mistake.

The following sentence was adapted:

Materials and methods, page 9, lines 186-188: “Next, generalized linear models were used to evaluate the association between hospitalisation and comorbidity on the one hand and the different measures of physical performance on the other.”

4i) Line 168: Stepwise methods or the variety of penalized regression modelling techniques are preferable modeling approaches to pulling out significant variables in a univariate analysis.

Thank you for this suggestion. We are aware of these approaches to pull out variables in a univariate analysis. We actually did perform a (backward) stepwise regression analysis. This description was added to the manuscript. We also agree with the reviewer that a penalized model technique might even be a more solid technique. This method (e.g. a ridge regression analysis), however, is complicated and often not understood by readers. We already performed such a penalized technique in previously published articles, and were confronted with the feedbacks that a simple statistical approach is often favoured over a complicated approach because it improves the transparency of the study (Vanden Wyngaert, K.; Van Craenenbroeck, A.H.; Holvoet, E.; Calders, P.; Van Biesen, W.; Eloot, S. Composite Uremic Load and Physical Performance in Hemodialysis Patients: A Cross-Sectional Study. Toxins 2020, 12, 135.). Therefore, we decided to stick with our backward regression analysis and to not use a penalized technique, especially because a stepwise regression method has been considered a valid method as well. Nevertheless, we would like to thank the reviewer for this comment because it improves the transparency of our manuscript.

Materials and methods, page 9, lines 182-186: “Also, the unique associations between measures of physical performance and survival were analysed via a univariate analysis. A multivariate stepwise backward regression model was used to examine which measure(s) of physical performance was/were most prominent in the association between physical performance and survival.”

4j) Line 169: Based on the results (comparing inferences in the univariate and multivariate models), it appears that more collinearity was a bigger issue than reported. The results do not make clear why the results are so divergent.

The reviewer is right to state that there seems to be some remaining collinearity present. Measures of physical function are related to each other, and collinearity between the different measures of physical performance is evident. However, we used a reliable tool to measure collinearity between de different measures of physical function and, subsequently, excluded the Dialysis Fall Risk Index (DFRI) from the analysis. To meet with the reviewer’s remark, we added the presence of a presumed collinearity to the limitations of the study.

The following sentence was added to the limitations:

Discussion, page 18, lines 375-377: “Second, although we excluded variables that caused multicollinearity into the statistical models, some collinearity between the remaining measures of physical function may still be present.”

4k) Lines 196/198: The change in mortality risk should be given in terms of the statistics they were based upon.

In order to improve the transparency of the results and to improve the consistency in reporting style, we adapted the following sentences in the revised version of the manuscript: 

Results, Page 11, line 218-222: “The risk of falls by DFRI was associated with an increased mortality risk (HR = 20.4, p=0.003). A walking distance of less than 298 meters (specificity = 0.583; sensitivity = 0.889, Fig. 1) was associated with an increased mortality risk (HR = 7.4, p=0.045).”

4l) Lines 200/202: The significance of 6MWT in this setting is surprising and difficult to believe, given how close it is to one. Standard errors or confidence intervals might help here.

Indeed, adding the confidence intervals would help the readers to interpret this result. The 95%CI was [0.994; 0.998]. I added this information to the manuscript.

The following information was added to the revised version of the manuscript:

Results, Page 11, line 224-226: “After adjusting for age, sex, comorbidity and dialysis vintage, each increase on the 6MWT (m) was associated with a 0.4% decrease in mortality risk (HR = 0.996, 95%CI [0.994; 0.998], p<0.001, Table 4).”

This information was also added to the abstract in the revised version of the manuscript:

Abstract, page 2, lines 36-38: “Each increase in 6MWT (m) corresponded with a 0.4% decrease in mortality risk (HR = 0.996, 95%CI [0.994; 0.998]).”

4m) Line 202: It is not clear how a hazard ratio of 0.996 translates into a 4% change in mortality risk.

Thank you for this comment. The reviewer is correct that HR = 0.996 does not translate in a 4% change in mortality risk. I adapted this information in the revised version of the manuscript.

Results, Page 11, line 224-226: “After adjusting for age, sex, comorbidity and dialysis vintage, each increase on the 6MWT (m) was associated with a 0.4% decrease in mortality risk (HR = 0.996, 95%CI [0.994; 0.998], p<0.001, Table 4).”

4n) Line 276-277: The robustness of the cut-off is not justified, since there is no obvious attempt to assess the factor 6MWT as it interacts with the covariates listed.

Thank you for this comment. Unfortunately, we have to disagree with the reviewer. We do not examine a causal relationship, but merely a predictive one. In a model examining causal relationships, confounding factors may bias the causal relationship between two variables. In a prediction-model, however, confounding is in essence not relevant. The question is not whether the 6MWT is associated with other variables that are associated with mortality, but whether the 6MWT itself is predictor of mortality. Therefore, we believe that the possible interactions of the 6MWT with other variables is not important to this analysis. 

REVIEWER #2

Overall, the idea of research is very interesting to be studied nowadays and paper is coherently developed. However, there are some comments and suggestions.

1) Abstract

1a) Page 2 line 21: add ''Aim'' before (To assess the association between different 22 measures of physical performance and survival in dialysis patients).

Thank you for this suggestion.

The following sentence was adapted:

Abstract, page 2, lines 21-22: “We aimed to assess the association between different measures of physical performance and survival in dialysis patients.”

1b) Page 2 line 26: write the full name of (FICSIT-4).

We adapted this sentence.

Abstract, page 2, line 26: “Muscle strength (quadriceps, handgrip strength, and sit-to-stand), exercise capacity (six-minute walking test, 6MWT) and the risk of falls (Dialysis Fall Index, Tinetti, and Frailty and Injuries: Cooperative Studies of Intervention Techniques) were measured at the time of inclusion.”

1c) Page 2 line 32: write the full name of (HR)

Thank you for this remark. We wrote HR in full the first time.

Abstract, page 2, lines 32-34: “All domains of physical performance were associated with mortality, with the highest hazards for an increased risk of falls (Hazard Ratio (HR) = 20.4, p=0.003) and poor exercise capacity (HR = 7.4, p<0.001).”

1d) Keywords: write it in alphabetical order

The order was adapted as suggested by the reviewer

Indexing Key Words, page 3, lines 45-46: “Cardiovascular disease; end-stage kidney disease; physical performance; morbidity; mortality”

2) Introduction

2a) Line 53 to 64, please insert the reference for this section

We would like to thank the reviewer for this remark. There are indeed two references lacking in the section. The study by Caspersen et al., is a key publication regarding terminology and should be added to the introduction.

The following references were added to the introduction:

Introduction, Page 4, lines 58-69: “Physical performance is an umbrella term that represents the different domains involved when performing physically challenging activities such as walking or daily chores. In general, it comprises the ability to make movements by producing muscle activity (muscle strength), the capacity to perform these movements continuously for a longer period (exercise capacity), and to exert the necessary control and coordination of movements. These domains can be assessed in a purely analytical way (e.g. quadriceps peak torque at knee 90° flexed, maximal muscle strength) or by simulating activities of daily living (e.g. getting up from a seating position, functional lower limb muscle strength) [4]. The fear of falls is an important cause of a sedentary lifestyle and physical deterioration as well. An increased risk of falls has a multifactorial aetiology in patients on haemodialysis, often involving both domains of physical performance and impaired balance and coordination; and assessment of the risk of falls should therefore be included in the physical screening of these patients [5].”

Reference: “4. Caspersen CJ, Powell KE, Christenson GM. Physical activity, exercise, and physical fitness: definitions and distinctions for health-related research. Public health reports (Washington, DC : 1974). 1985;100(2):126-31. PubMed PMID: 3920711.”

Reference: “5. Kono K, Nishida Y, Yabe H, Moriyama Y, Mori T, Shiraki R, et al. Development and validation of a Fall Risk Assessment Index for dialysis patients. Clinical and experimental nephrology. 2018;22(1):167-72. Epub 2017/06/22. doi: 10.1007/s10157-017-1431-8. PubMed PMID: 28634773.”

3) Materials and methods

3a) Study design is not mentioned

The study design was indeed lacking. This study reports longitudinal data. This information was added to the Materials and Methods.

Materials and methods, page 5, lines 92-94: “This longitudinal study was part of a project to identify potential (predicting) biomarkers based on functional capacity, nutritional status and/or QoL in patients on dialysis (registration number on clinicaltrial.gov: NCT03910426).”

3b) Correct the title on page 4 line 76 (Materials & methods) to (Materials and methods)

The title was adapted in the revised version of the manuscript.

3c) Page 5 line 103: please clarify the (Quadriceps peak torque) test protocol as did you examine eccentric or concentric Quadriceps peak torque, the knee at angular velocity.

Thank you for this pertinent remark. A description of the quadriceps peak torque assessments is indeed lacking. Patients had to be in a sitting position with 90° knee and 90° hip flexion. The hand-held dynamometer was placed on the anterior aspect of the tibia of the dominant leg, proximal to the ankle joint. The participants had to perform a voluntary maximal isometric contraction for five seconds against the dynamometer, which was fixated by the physiotherapist which applied enough resistance so movement of the limb was avoided and maximal strength was assessed (Mentiplay BF, Perraton LG, Bower KJ, Adair B, Pua Y-H, Williams GP, et al. Assessment of Lower Limb Muscle Strength and Power Using Hand-Held and Fixed Dynamometry: A reliability and Validity Study. 2015). The leg was tested three times with one minute rest in-between the assessments. The best out of the three attempts was noted. Verbal encouragement was given. A short description of the protocol was added to the revised version of the manuscript.

Materials and methods, page 6, lines 118-119: “Quadriceps strength was measured by maximal isometric contraction and patients had to be in a sitting position with 90° flexion in both knee and hip.”

3d) Validity and reliability of the used instruments used in this study should be clarified

Thank you for this methodological suggestion. We added the validity and reliability of the used instruments to materials and methods. 

The following information was added to the revised version of the manuscript:

Materials and methods, page 6, lines 113-118: “Maximal muscle strength. Quadriceps peak torque (Microfet; Biometrics, Almere, the Netherlands) (Intraclass Correlation Coefficient (ICC) ranging between 0.76 and 0.96) [13, 14] and handgrip strength (JAMAR Hydraulic Hand Dynamometer; Patterson, Nottinghamshire, United Kingdom) (ICC = 0.93) [15] were measured by hand-held dynamometers according to the guidelines of Bohannon [16] and American Society of Hand Therapists [17] respectively.”

Materials and methods, page 7, lines 123-125: “The five repetition Sit-to-Stand test (STS) was used to examine functional lower limb muscle strength and was expressed as the time patients needed to stand up 5 times from a seating position (ICC = 0.97) [19, 20].”

Materials and methods, page 7, lines 129-131: “The six-minute walking test (6MWT) is considered the gold standard for assessing functional exercise capacity and was performed following the American Thoracic Society guidelines (ICC > 0.90) [22, 23]”

Materials and methods, page 7, lines 140-142: “Next, the Tinetti (ICC > 0.8) [25] and FICSIT tools (Frailty and Injuries Cooperative Studies of Intervention Technique) (ICC = 0.79) [26] were used to examine gait dysfunctions and static balance [27].”

The following references were also added to the revised version of the manuscript:

References: “13. Mentiplay BF, Perraton LG, Bower KJ, Adair B, Pua YH, Williams GP, et al. Assessment of Lower Limb Muscle Strength and Power Using Hand-Held and Fixed Dynamometry: A Reliability and Validity Study. PloS one. 2015;10(10):e0140822. Epub 2015/10/29. doi: 10.1371/journal.pone.0140822. PubMed PMID: 26509265; PubMed Central PMCID: PMCPMC4624940.”

References: “14. Vanpee G, Hermans G, Segers J, Gosselink R. Assessment of limb muscle strength in critically ill patients: a systematic review. Critical care medicine. 2014;42(3):701-11. Epub 2013/11/10. doi: 10.1097/ccm.0000000000000030. PubMed PMID: 24201180.”

References: “15. Hermans G, Clerckx B, Vanhullebusch T, Segers J, Vanpee G, Robbeets C, et al. Interobserver agreement of Medical Research Council sum-score and handgrip strength in the intensive care unit. Muscle & nerve. 2012;45(1):18-25. Epub 2011/12/23. doi: 10.1002/mus.22219. PubMed PMID: 22190301.”

References: “20. Ng SS, Kwong PW, Chau MS, Luk IC, Wan SS, Fong SS. Effect of arm position and foot placement on the five times sit-to-stand test completion times of female adults older than 50 years of age. J Phys Ther Sci. 2015;27(6):1755-9. Epub 2015/07/17. doi:”

References: “23. Demers C, McKelvie RS, Negassa A, Yusuf S. Reliability, validity, and responsiveness of the six-minute walk test in patients with heart failure. American heart journal. 2001;142(4):698-703. Epub 2001/10/02. doi: 10.1067/mhj.2001.118468. PubMed PMID: 11579362.”

References: “25. Kegelmeyer DA, Kloos AD, Thomas KM, Kostyk SK. Reliability and validity of the Tinetti Mobility Test for individuals with Parkinson disease. Physical therapy. 2007;87(10):1369-78. Epub 2007/08/09. doi: 10.2522/ptj.20070007. PubMed PMID: 17684089.”

References: “26. Blankevoort CG, van Heuvelen MJ, Scherder EJ. Reliability of six physical performance tests in older people with dementia. Physical therapy. 2013;93(1):69-78. Epub 2012/09/15. doi: 10.2522/ptj.20110164. PubMed PMID: 22976448.”

3e) Page 7 line 163 the authors mentioned the reference Fine and Gray [23, 24]. However 23 and are two different references. Correct

Thank you for this remark. We corrected this mistake. 

The following sentence was adapted:

Materials and methods, page 9, lines 178-180: “We applied a competing risk survival analysis (competing event: transplantation) to examine the association between the different measures of physical performance and mortality as described by Fine and Gray [31]”

3f) Delete the word abbreviation bellow all tables

The term ‘abbreviation’ was deleted below all tables.

4) Funding

4a) Page 17 line 372 and 373 include funding and conflicts of interest. Please write in separate

The statements were adapted.

The following information was adapted in the revised version of the manuscript:

Discussion, page 19, lines 396-401: “

Conflict of interest

There are no conflicts of interest.

Funding

This research did not receive any specific grant from funding agencies in the public, commercial, or not-for-profit sectors.”

5) References

5a) Update some references is required

Thank you for this remark. All references were checked and updated (including the newly added references).

---

## [Decision Letter · Decision Letter 1]

21 Dec 2021

PONE-D-20-37730R1The importance of physical performance in the assessment of patients on haemodialysis: a survival analysisPLOS ONE

Dear Dr. Vanden Wyngaert,

Thank you for submitting your manuscript to PLOS ONE. After careful consideration, we feel that it has merit but does not fully meet PLOS ONE’s publication criteria as it currently stands. Therefore, we invite you to submit a revised version of the manuscript that addresses the points raised during the review process. One of the reviewers has raised addressable concerns with some of the statistical analyses. Statistical methods need to be addressed to meet PLOS ONE publication criteria, specifically item #3.

We look forward to receiving your revised manuscript.

Kind regards,

Melissa M Markofski

Academic Editor

PLOS ONE

Journal Requirements:

Reviewers' comments:

Reviewer's Responses to Questions

**Comments to the Author**

1. If the authors have adequately addressed your comments raised in a previous round of review and you feel that this manuscript is now acceptable for publication, you may indicate that here to bypass the “Comments to the Author” section, enter your conflict of interest statement in the “Confidential to Editor” section, and submit your "Accept" recommendation.

Reviewer #1: (No Response)

Reviewer #2: All comments have been addressed

2. Is the manuscript technically sound, and do the data support the conclusions?

Reviewer #1: Partly

Reviewer #2: Yes

3. Has the statistical analysis been performed appropriately and rigorously? 

Reviewer #1: No

Reviewer #2: Yes

4. Have the authors made all data underlying the findings in their manuscript fully available?

Reviewer #1: No

Reviewer #2: Yes

5. Is the manuscript presented in an intelligible fashion and written in standard English?

Reviewer #1: Yes

Reviewer #2: Yes

6. Review Comments to the Author

Reviewer #1: The current version of this article represents a notable improvement over the initial version. However, concerns do remain.

The term "competing risk" should be modified to be "competing risks".

A description of the rate of censoring should be provided. How many subjects dropped out early? What was the rate per annum? What was the median/maximum time of censoring?

The use of a cutoff for 6MWT is confusing. The cutoff was (seemingly) based on a categorical analysis of overall mortality, as opposed to a benchmark level of mortality. Possible bias a result of censoring was not considered. Using mortality as an outcome both for finding a cutoff and assessing the hazard ratio risk brings bias both in to the estimation of the hazard ratio risk and the inference of a such a risk. Unless it is absolutely justified otherwise, the cutoff should be based on an optimization in the survival modeling space, based on a performance metric such as concordance. Additionally, the sensitivity of the optimal cutoff to population subset selection and/or variable should be included.

The number of units reported in a continuous factor, as it affects a survival outcome, should correspond to a meaningful effect on mortality. As such the effect on mortality with respect to 10 or more meters increase in 6MWT should be reported. The resulting hazard ratio (about 0.96) would have more meaning clinically.

Line 88: "Predicting" should be "predictive".

When reporting a variable, if the standard deviation exceeds the mean, there is a strong indication of a skew in the distribution. As a result, the 25% and 75% quantiles should be reported. This applies specifically to the variables "Dialysis vintage" and "Davies comorbidity score" in Table 1.

More description is needed for the randomization scheme and the variable selection procedure. For the randomization scheme, how was a particular randomization assigned to each patient? Was balance accounted for or assessed? Was the effect of ordering considered in the statistical modeling? With respect to variable selection, what criterion (e.g., AIC or BIC) was used to choose the appropriate model? It appears that the variable selection procedure did not actually take out any variables for the final multivariable models; is that the case?

Table 3 should be described more specifically. What subset of the population was used? What was the criterion of selection? How large was that subset?

Since modeling using both the cutoff version of 6MWT and the continuous version of 6MWT were used in the modeling, the write-up in the Results of the use of these variables should be more careful. In particular, the description on lines 214-221 jumps between the two characterizations in a way that could be confusing to the reader.

Reviewer #2: Comments to authors

Thank you for your effort to revise your manuscript. You have done great effort and responded to suggested comments. I have no further comments on your work.

7. PLOS authors have the option to publish the peer review history of their article (what does this mean?). If published, this will include your full peer review and any attached files.

Reviewer #1: No

Reviewer #2: No

---

## [Author Response · Author response to Decision Letter 1]

5 Jan 2022

REVIEWER #1

The current version of this article represents a notable improvement over the initial version. However, concerns do remain.

We thank the reviewer for his/her constructive criticism and observation of our effort to improve the manuscript according to the provided suggestions.

1) The term "competing risk" should be modified to be "competing risks".

Thank you for this linguistic suggestion.

The following sentences were adapted in the revised version of the manuscript:

Abstract, page 2, lines 26-27: “Data were analysed by general linear models and competing risks survival hazard models.”

Materials and methods, page 8, lines 145-147: “Transplantation during the study period was no exclusion criteria in the final survival-analyses as this would introduce censoring bias [29], they were however considered as a competing risks of survival and therefore censored via the appropriate analysis.”

Materials and methods, page 9, lines 175-177x: “We applied a competing risks survival analysis (competing event: transplantation) to examine the association between the different measures of physical performance and mortality as described by Fine and Gray [31].”

Results, page 11, lines 216-217: “All domains of physical performance were associated with survival in the univariate competing risks regression models (Tables 2 and 3).”

The following titles of Tables 2-5 were adapted in the revised version of the manuscript:

Results, page 12, line 231: “Table 2. Unadjusted competing risks regression model for mortality.”

Results, page 12, lines 236-237: “Table 3. Unadjusted competing risks regression model for mortality by the presence of impairments in physical performance.”

Results, page 12, line 243: “Table 4. Adjusted competing risks regression model for mortality.”

Results, page 13, lines 248-249: “Table 5. Adjusted multivariate competing risks regression model by the presence of impairments in physical performance.”

2) A description of the rate of censoring should be provided. How many subjects dropped out early? What was the rate per annum? What was the median/maximum time of censoring?

Based on your suggestion, we added extra data on the censoring of patients. The rate of censoring (i.e. censored population over the population with no observed event) was 18%. We added this information to the revised version of the manuscript. No patients dropped out early and, consequently, there was no lost-to-follow-up. We added this information as well to the manuscript. Additionally, the mortality rate per annum was 35.6% (16 patients), 33.3% (15 patients), 13.3% (6 patients), and 17.8% (8 patients) for the 1st to 4th annum. With respect to the competing risks, 6, 4, and 3 patients were transplanted in the first, second and third annum respectively. Also the minimum and maximum time of censoring was added (i.e. minimum time of censoring 49 days; maximum time 796 days).

The following information was added to the revised version of the manuscript:

Results, page 10, lines 196-203: “During an observation period of 33 months on average (ranging between 30 and 45 months), 45 subjects (38.5%) died and 13 patients (11.1%) were transplanted, which resulted in a mortality rate of 15% and rate of censoring of 18%. No patients dropped out during the observation period. Sixteen, fifteen, six, and eight patients died in the first, second, third, and fourth annum, which corresponds with a relative ratio of 35.6%, 33.3%, 13.3%, and 17.8%. With respect to the competing risks, six, four, and three patients were transplanted in the first, second and third annum respectively, resulting in a medium censoring time of 425 days, [minimum 49 days; maximum 796 days].”

3) The use of a cutoff for 6MWT is confusing. The cutoff was (seemingly) based on a categorical analysis of overall mortality, as opposed to a benchmark level of mortality. Possible bias a result of censoring was not considered. Using mortality as an outcome both for finding a cutoff and assessing the hazard ratio risk brings bias both in to the estimation of the hazard ratio risk and the inference of a such a risk. Unless it is absolutely justified otherwise, the cutoff should be based on an optimization in the survival modeling space, based on a performance metric such as concordance. Additionally, the sensitivity of the optimal cutoff to population subset selection and/or variable should be included.

We would like to thank the reviewer for this remark because the statistical methods were indeed inadequately described. Various methods are reported and used in the literature to select a cut-off point which defines two groups with different risks for a continuously measured marker (e.g. the 6MWT). Only two of them are based on the maximization of the true classification rate, i.e. the Youden’s Index and the concordance probability (Unal I. (2017). Defining an Optimal Cut-Point Value in ROC Analysis: An Alternative Approach. Computational and mathematical methods in medicine, 2017, 3762651. & Kim SJ, Myong JP, Suh H, Lee KE, Youn YK. Optimal Cutoff Age for Predicting Mortality Associated with Differentiated Thyroid Cancer. PLoS One. 2015;10(6):e0130848. Published 2015 Jun 23.). The reviewer is right that other methods, e.g. the point closest to the ROC curve, are inferior and inappropriate to the above mentioned methods and induce a considerable risk of bias (Perkins NJ, Schisterman EF. The inconsistency of "optimal" cutpoints obtained using two criteria based on the receiver operating characteristic curve. Am J Epidemiol. 2006;163(7):670-675.). In our manuscript, we only used the ROC-curve to identify potentially relevant outcome measures (AUC above 0.7) (Mandrekar JN. Receiver operating characteristic curve in diagnostic test assessment. J Thorac Oncol. 2010 Sep;5(9):1315-6.). Subsequently, we used the Youden’s index to identify the most optimal cut-off point for our population. In the previous versions of the manuscript, we wrongly reported that a ROC-analysis (suggesting the ‘closest point to 0,1’-method) was used to determine a cut-off point for 6MWT; this statement was incorrected. We corrected this information. 

The following sentences were adapted in the revised version of the manuscript:

Results, pages 8-9, lines 166-171: “We performed a receiver operating characteristic (ROC) analysis to identify relevant outcome measures (i.e. measures with an area under the curve (AUC) above 0.7) Subsequently, based on the Youden’s index, a 298 m cut-off point (specificity = 0.583; sensitivity = 0.889) on the 6MWT test was established and used in the survival analysis (Fig 1).”

We also agree with the reviewer that some bias could be present because we used data to calculate both a cut-off point and hazard ratio’s. Additionally, residual bias may be present due to censoring. Therefore, we added these concerns to the limitations of the manuscript. Additionally, we also included a statement in future research perspectives on the recommendation to validate and examine this cut-off point in other cohorts of patients on haemodialysis.

The following sentences were adapted in the revised version of the manuscript:

Discussion, page 18, lines 378-382: “Fourth, mortality data were used to calculate both a cut-off point and hazard ratio’s, potentially bringing some bias to the analysis. In addition, bias due to censoring might still be present despite the use of a competing risks analysis. Therefore, our data should be interpreted with caution.”

Conclusions and guidelines for further research, page 19, lines 394-396: “Additionally future research should try to validate and examine the cut-off point in other cohorts of haemodialysis patients as well as to examine the true clinical importance of the associations reported in the present study.”

4) The number of units reported in a continuous factor, as it affects a survival outcome, should correspond to a meaningful effect on mortality. As such the effect on mortality with respect to 10 or more meters increase in 6MWT should be reported. The resulting hazard ratio (about 0.96) would have more meaning clinically.

We agree with the reviewer that the establishment of a Minimal Clinically Important Difference (MCID) would be interesting. Such a MCID would be of added value to clinicians in their interpretation of the 6MWT as well as the change of 6MWT over time. However, such a MCID is preferably examined via a longitudinal study with multiple assessments over time (Crosby RD, Kolotkin RL, Williams GR. Defining clinically meaningful change in health-related quality of life. J Clin Epidemiol. 2003 May;56(5):395-407.). Therefore, our data is not appropriate to be used in a MCID-analysis. Nevertheless, our data indicate that measures of physical function are associated with mortality and future research should examine the true (clinical) importance of this association. To meet with the reviewers suggestion, we added a statement to Future Research Perspectives.

The following sentence was added to the revised version of the manuscript:

Conclusions and guidelines for further research, page 19, lines 394-396: “Additionally future research should try to validate and examine the cut-off point in other cohorts of haemodialysis patients as well as to examine the true clinical importance of the associations reported in the present study.”

5) Line 88: "Predicting" should be "predictive".

We corrected the sentence.

The following sentence was adapted in the revised version of Materials and methods:

Materials and methods, page 5, lines 87-89: “This longitudinal study was part of a project to identify potential (predictive) biomarkers based on functional capacity, nutritional status and/or QoL in patients on dialysis (registration number on clinicaltrial.gov: NCT03910426).”

6) When reporting a variable, if the standard deviation exceeds the mean, there is a strong indication of a skew in the distribution. As a result, the 25% and 75% quantiles should be reported. This applies specifically to the variables "Dialysis vintage" and "Davies comorbidity score" in Table 1.

Thank you for this remark. We performed the requested adaptations.

The following information was adapted in the revised version of the manuscript:

Results, pages 10-11, lines 210-213: “

Table 1. Patient characteristics.

Variable n = 117

Dialysis vintage (months) 36 [11; 55]

Davies comorbidity score (0-7) 2 [1; 3]

Data are presented as mean ± standard deviation, as number (percentage) and as mean [25th; 75th] quantiles. BMI, body mass index; CVD, cardiovascular disease; VAS, visual analogue scale; 6MWT, 6-minute walking test”

7) More description is needed for the randomization scheme and the variable selection procedure. For the randomization scheme, how was a particular randomization assigned to each patient? Was balance accounted for or assessed? Was the effect of ordering considered in the statistical modeling? With respect to variable selection, what criterion (e.g., AIC or BIC) was used to choose the appropriate model? It appears that the variable selection procedure did not actually take out any variables for the final multivariable models; is that the case?

We randomized the sequence of physical examinations by using six opaque envelopes. Each envelope containing an assessment of physical function, i.e. (1) quadriceps strength, (2) handgrip strength, (3) sit-to-stand test, (4) six-minute walking test, (5) Tinetti test, and (6) the FICSIT. The patients were asked to order the opaque envelopes randomly and the tests were performed as such. 

This information was added to the revised version of the manuscript:

Materials and methods, page 6, lines 96-99: “The sequence of physical examinations was randomized using six opaque envelopes (one for each assessment of physical function, i.e. quadriceps strength, handgrip strength, sit-to-stand test, six-minute walking test, Tinetti test, and the FICSIT) and the patients were asked to randomly order the envelopes.”

Balance was assessed via the FICSIT test. The ‘Frailty and Injuries: Cooperative Studies of Intervention Techniques-4’ is a static balance test commonly used for a variety of elderly populations (Rossiter-Fornoff J, Wolf S, Wolfson L, Buchner D. A cross-sectional validation study of the FICSIT common data base static balance measures. Frailty and Injuries: Cooperative Studies of Intervention Techniques. Journals of Gerontology Series A1995. p. 291-7 & Blankevoort CG, van Heuvelen MJ, Scherder EJ. Reliability of six physical performance tests in older people with dementia. Phys Ther. 2013;93(1):69-78). To my knowledge, the scale has not been studied in patients on haemodialysis and no clear cut-off points were reported in the literature. Patients were asked to take a sequence of 7 positions with progressive difficulty but without the use of any assistive devices, the positions were: (1-2) parallel stance with feet closely together (eyes open and closed), (3-4) semi-tandem stance (eyes open and closed), (5-6) tandem stance (eyes open and closed) and (7) unipedal stance with the preferred leg. Patients were scored on each position to a 5-point scale with a total maximum score of 28, based on test-time (up to 10 seconds) and need for feedback or aid. If patients were not able to complete a subtest, the assessment of the FICSIT-4 stopped (Rossiter-Fornoff J, Wolf S, Wolfson L, Buchner D. A cross-sectional validation study of the FICSIT common data base static balance measures. Frailty and Injuries: Cooperative Studies of Intervention Techniques. Journals of Gerontology Series A1995. p. 291-7).

The effect of ordering was not examined in the statistical analysis. Only measures that showed significant associations in the univariate model were introduced in the multivariate model. No stepwise method was used in the present study. This was wrongfully reported in Materials and methods. We apologize for this mistake, which happened because a stepwise backward regression model was used previously in other published articles. Although a stepwise regression model is superior to the method that we used in the present study, we were not able to combine this statistical method with the competing risks regression model. We added this limitation to the discussion of the manuscript.

The following sentence was added to the revised Discussion:

Discussion, page 18, lines 381-382: “Fifth, the variable selection in the multivariate analysis was based on the univariate analysis and not on a stepwise selection model.”

8) Table 3 should be described more specifically. What subset of the population was used? What was the criterion of selection? How large was that subset?

Thank you for this suggestion to make the manuscript more transparent. The same criteria were used in Table 3 as in Table 1 to identify impairments in measures of physical function of patients. More information of the subset of the population was added to Table 3 (i.e. the ratio of patients with and without impaired physical function).

The following information was added to Table 3:

Results, page 12, lines 236-237: “

Table 3. Unadjusted competing risks regression model for mortality by the presence of impairments in physical performance.

 Mortality (event = death)

Impairments (impaired (n); not impaired (n)) Univariate model Multivariate model

 HR p-value HR p-value

Quadriceps strength (108; 9) 5.291 0.045 2.611 0.293

Handgrip strength (43; 74) 2.433 0.003 1.721 0.079

Sit-to-Stand (81; 36) 4.566 0.001 1.403 0.573

Tinetti (58; 59) 3.215 <0.001 0.001 0.778

DFRI (87; 30) 20.408 0.003 / NA

6MWT (80; 37) 7.407 <0.001 5.882 0.002

”

9) Since modeling using both the cutoff version of 6MWT and the continuous version of 6MWT were used in the modeling, the write-up in the Results of the use of these variables should be more careful. In particular, the description on lines 214-221 jumps between the two characterizations in a way that could be confusing to the reader.

We improved the transparency of the manuscript by adding more details on the type of the data in this section.

The following sentences were adapted in the revised version of the manuscript:

Results, page 11, lines 221-229: “A walking distance of less than 298 meters (specificity = 0.583; sensitivity = 0.889, Fig 1) was associated with an increased mortality risk (HR = 7.4, p=0.045). Only exercise capacity as a continuous variable was retained as a predictor of survival in the multivariate analyses (6MWT (m): HR = 0.997, p=0.005; impaired 6MWT: HR = 5.880, p=0.002). After adjusting for age, sex, comorbidity and dialysis vintage, each increase on the 6MWT (m) was associated with a 0.4% decrease in mortality risk (HR = 0.996, 95%CI [0.994; 0.998], p<0.001, Table 4). The results of the adjusted survival analysis for the 298 meters walking distance cut-off point (i.e. the categorical analysis of exercise capacity) were similar to the results of the unadjusted analysis (HR = 7.576, p<0.001, Table 5).”

REVIEWER #2

Thank you for your effort to revise your manuscript. You have done great effort and responded to suggested comments. I have no further comments on your work.

Thank you for the constructive remarks.

---

## [Decision Letter · Decision Letter 2]

2 Feb 2022

PONE-D-20-37730R2The importance of physical performance in the assessment of patients on haemodialysis: a survival analysisPLOS ONE

Dear Dr. Vanden Wyngaert,

Thank you for submitting your manuscript to PLOS ONE. After careful consideration, we feel that it has merit but does not fully meet PLOS ONE’s publication criteria as it currently stands. Therefore, we invite you to submit a revised version of the manuscript that addresses the points raised during the review process. The majority of the previous comments were addressed, but there are still some concerns. 

We look forward to receiving your revised manuscript.

Kind regards,

Melissa M Markofski

Academic Editor

PLOS ONE

Journal Requirements:

Reviewers' comments:

Reviewer's Responses to Questions

**Comments to the Author**

1. If the authors have adequately addressed your comments raised in a previous round of review and you feel that this manuscript is now acceptable for publication, you may indicate that here to bypass the “Comments to the Author” section, enter your conflict of interest statement in the “Confidential to Editor” section, and submit your "Accept" recommendation.

Reviewer #1: (No Response)

2. Is the manuscript technically sound, and do the data support the conclusions?

Reviewer #1: Partly

3. Has the statistical analysis been performed appropriately and rigorously? 

Reviewer #1: I Don't Know

4. Have the authors made all data underlying the findings in their manuscript fully available?

Reviewer #1: No

5. Is the manuscript presented in an intelligible fashion and written in standard English?

Reviewer #1: Yes

6. Review Comments to the Author

Reviewer #1: The authors are to be congratulated on addressing a great many issues related to the technical and data analytic aspects of this article.

However, several important issues have not been fully addressed.

Lines 167-169 Receiver operating character analysis and specificity/sensitivity analyses are diagnostic functions of a generalized regression model. From the text, it can be inferred which predictors -- 6MWT, for instance -- are being used, but the authors have still not specified which outcome is associated with this modeling. Is it death? What categorical outcome is being used to generate these results?

Several issues remain with the Statistical Analysis section.

163 For which reported outcome (in the text) was the chi-square test used?

Lines 255-284 For the comorbidity analysis, the outcomes seem to be ordinal categories based on the Davies comorbidity index. Was this the case? Please specificy.

Additionally, please provide a more detailed description of the modeling that was done for this analysis. A generalized linear model (or a "general linear model") is not a sufficient description.

7. PLOS authors have the option to publish the peer review history of their article (what does this mean?). If published, this will include your full peer review and any attached files.

Reviewer #1: No

---

## [Author Response · Author response to Decision Letter 2]

4 Feb 2022

REVIEWER #1

The authors are to be congratulated on addressing a great many issues related to the technical and data analytic aspects of this article. However, several important issues have not been fully addressed.

Thank you for this comment. We are happy that most of the items were sufficiently addressed. 

1) Lines 167-169 Receiver operating character analysis and specificity/sensitivity analyses are diagnostic functions of a generalized regression model. From the text, it can be inferred which predictors -- 6MWT, for instance -- are being used, but the authors have still not specified which outcome is associated with this modeling. Is it death? What categorical outcome is being used to generate these results?

Thank you for clarifying this previous comment. The categorical outcome measure was indeed mortality (death). Because this information was lacking, we added this information to the sentence.

The following sentence was adapted in the revised version of the manuscript:

Materials and methods, page 9, lines 167-169: “We performed a receiver operating characteristic (ROC) analysis to identify relevant outcome measures (i.e. measures with an area under the curve (AUC) above 0.7) with regard to mortality.”

2) Several issues remain with the Statistical Analysis section. 163 For which reported outcome (in the text) was the chi-square test used?

Thank you for addressing this error. If we would have examined the differences in incidence of death between groups, we would have used the chi-square test. However, the chi-square to examine such a difference was indeed not used. We deleted this sentence.

4) Lines 255-284 For the comorbidity analysis, the outcomes seem to be ordinal categories based on the Davies comorbidity index. Was this the case? Please specificy.

The reviewer is correct. The Davies comorbidity index is an ordinal variable, which can identify patients with a low, medium or high mortality risk. However, the results reported in Tables 6 and 7 are based on the Davies Comorbidity scale (0-7), which is the sum of active (relevant) comorbid conditions in a patient. We clarified the results by adding this information.

The following information was added to the revised version of results:

Results, page 13, lines 262-265: “Similar to the survival analysis, all domains of physical performance were associated with the degree of morbidity based on the Davies comorbidity scale (0-7), albeit only the 6MWT remained relevant in the multivariate analysis (R2 = 0.145, p=0.012, Table 6).”

Results, page 13, lines 265-266: “Adjusted for age, sex and dialysis vintage, the 6MWT explained 20.8% of the variance in the Davies comorbidity scale (p<0.001, Table 7).”

Results, page 14, lines 268-269: “Table 6. Unadjusted general linear model of the different measures of physical performance on comorbidity (Davies comorbidity scale 0-7).”

Results, page 14, lines 273-274: “Table 7. Adjusted general linear model of the 6MWT on comorbidity (Davies comorbidity scale 0-7).”

5) Additionally, please provide a more detailed description of the modeling that was done for this analysis. A generalized linear model (or a "general linear model") is not a sufficient description.

Thank you for this remark. Because we used the comorbidity scale and not the index, we considered the scale as a continuous variable and, therefore, used a linear model. The reviewer is correct that a different model should have been used when the index (low, medium, high risk classification) was used. Nevertheless, we are of the opinion that a general linear model was appropriate to examine the association between comorbidity (as a continuous variable) and measures of physical performance.

---

## [Decision Letter · Decision Letter 3]

30 Mar 2022

PONE-D-20-37730R3The importance of physical performance in the assessment of patients on haemodialysis: a survival analysisPLOS ONE

Dear Dr. Vanden Wyngaert,

Thank you for submitting your manuscript to PLOS ONE. After careful consideration, we feel that it has merit but does not fully meet PLOS ONE’s publication criteria as it currently stands. Therefore, we invite you to submit a revised version of the manuscript that addresses the points raised during the review process.

 There is one minor clarification request by the reviewer. Unless there is more to add by the authors, making the requested clarification will not have to go back to reviewers. I can quickly check it when it is resubmitted. 

We look forward to receiving your revised manuscript.

Kind regards,

Melissa M Markofski

Academic Editor

PLOS ONE

Journal Requirements:

Reviewers' comments:

Reviewer's Responses to Questions

**Comments to the Author**

1. If the authors have adequately addressed your comments raised in a previous round of review and you feel that this manuscript is now acceptable for publication, you may indicate that here to bypass the “Comments to the Author” section, enter your conflict of interest statement in the “Confidential to Editor” section, and submit your "Accept" recommendation.

Reviewer #1: (No Response)

2. Is the manuscript technically sound, and do the data support the conclusions?

Reviewer #1: Yes

3. Has the statistical analysis been performed appropriately and rigorously? 

Reviewer #1: Yes

4. Have the authors made all data underlying the findings in their manuscript fully available?

Reviewer #1: No

5. Is the manuscript presented in an intelligible fashion and written in standard English?

Reviewer #1: Yes

6. Review Comments to the Author

Reviewer #1: The authors are to be congratulated on their responses to this and previous revisions. The responses and subsequent revisions have done much to improve the clarity of the analysis description.

There continues to be one minor comment:

Instead of the term "general linear model" (which can easily be confused with a "generalized linear model"), the term "least squares linear regression model" is preferred. Unless there is some reason not described in previous revisions, the term "general linear model" should be avoided in favor of preferred standard terminology.

7. PLOS authors have the option to publish the peer review history of their article (what does this mean?). If published, this will include your full peer review and any attached files.

Reviewer #1: No

---

## [Author Response · Author response to Decision Letter 3]

5 Apr 2022

The authors are to be congratulated on their responses to this and previous revisions. The responses and subsequent revisions have done much to improve the clarity of the analysis description. 

Thank you for the constructive suggestions and feedback.

There continues to be one minor comment: Instead of the term "general linear model" (which can easily be confused with a "generalized linear model"), the term "least squares linear regression model" is preferred. Unless there is some reason not described in previous revisions, the term "general linear model" should be avoided in favor of preferred standard terminology.

Thank you for this suggestion, which will improve the transparency of our manuscript. We adapted the term “general linear model” as requested.

The following sentences were adapted in the revised version of the manuscript:

Abstract, page 2, lines 26-28: “Data were analysed by least squares linear regression models and competing risks survival hazard models.”

Materials and methods, page 9, lines 182-184: “Next, least squares linear regression models were used to evaluate the association between hospitalisation and comorbidity on the one hand and the different measures of physical performance on the other.”

Results, page 13, lines 267-269: “Table 6. Unadjusted least squares linear regression model of the different measures of physical performance on comorbidity (Davies comorbidity scale 0-7).”

Results, page 14, lines 273-274: “Table 7. Adjusted least squares linear regression model of the 6MWT on comorbidity (Davies comorbidity scale 0-7).”

Results, page 14, lines 283-284: “Table 8. Least squares linear regression model of the different measures of physical performance on 1-year hospitalisation (days/year).

---

## [Editor Report · Decision Letter 4]

25 Apr 2022

The importance of physical performance in the assessment of patients on haemodialysis: a survival analysis

PONE-D-20-37730R4

Dear Dr. Vanden Wyngaert,

We’re pleased to inform you that your manuscript has been judged scientifically suitable for publication and will be formally accepted for publication once it meets all outstanding technical requirements.

Kind regards,

Melissa M Markofski

Academic Editor

PLOS ONE
---

## [Editor Report · Acceptance letter]

11 May 2022

PONE-D-20-37730R4 

The importance of physical performance in the assessment of patients on haemodialysis: a survival analysis 

Dear Dr. Vanden Wyngaert:

I'm pleased to inform you that your manuscript has been deemed suitable for publication in PLOS ONE. Congratulations! Your manuscript is now with our production department. 

Kind regards, 

on behalf of

Dr. Melissa M Markofski 

Academic Editor

PLOS ONE